# Characterizing the ability of LLMs to recapitulate Americans' distributional responses to public opinion polling questions across political issues

## Abstract

Traditional survey-based political issue polling is becoming less tractable due to increasing costs and risk of bias associated with growing non-response rates and declining coverage of key demographic groups. With researchers and pollsters seeking alternatives, Large Language Models have drawn attention for their potential to augment human population studies in polling contexts. We propose and implement a new framework for anticipating human responses on multiple-choice political issue polling questions by directly prompting an LLM to predict a distribution of responses. By comparison to a large and high quality issue poll of the US population, the Cooperative Election Study, we evaluate how the accuracy of this framework varies across a range of demographics and questions on a variety of topics, as well as how this framework compares to previously proposed frameworks where LLMs are repeatedly queried to simulate individual respondents. We find the proposed framework consistently exhibits more accurate predictions than individual querying at significantly lower cost. In addition, we find the performance of the proposed framework varies much more systematically and predictably across demographics and questions, making it possible for those performing AI polling to better anticipate model performance using only information available before a query is issued.

## 1 Introduction

Political issue polling is a billion-dollar industry that plays a pivotal role in the functioning of modern representative democratic systems. It helps shape policy at all levels of government. Candidates rely on it for feedback on their stated positions. Policymakers rely on it to understand the sentiments of the populace. Elected officials rely on it for feedback on implemented policies.

However, dramatic increases in non-response rates (Luiten et al., 2020) and the decline of effective, systematic methods to reach all respondents equally (Berinsky, 2017) threaten the industry's ability to produce accurate results reflective of target populations. Pollsters have responded with a variety of tactics. They continuously change outreach methods, aggregating responses derived from multiple different polling strategies (Kennedy et al., 2023). They use statistical methods to ensure that collected responses reflect the sentiments of the population (Berinsky, 2017). At the same time, they investigate other avenues for gauging public opinion; alternatives to the time-intensive, costly, and increasingly inaccurate traditional methodologies of political issue polling.

For more than a decade, researchers and pollsters have turned to model-based "listening" approaches, collating public opinion from vast pools of online resources containing written sentiments—such as social media platforms—as an alternative to polling for measuring public sentiment (Larsen & Fazekas, 2021; Murphy et al., 2014). These foundational explorations have paved the way for new techniques to infer public opinion without requiring new data collection or concerted efforts to induce sentiments from individuals via surveys or polls.

Large Language Models (LLMs) have the potential to vastly augment such alternative polling strategies (Sanders et al., 2023; Argyle et al., 2023; Pachot & Petit, 2024; Cerina & Rouméas, 2025). LLMs trained upon vast swaths of content from the internet, including the voices and opinions of individuals on social media and online forums, have become adept at a wide variety of tasks that require and demonstrate a mimicry of human facilities. They are able to generate written text nearly indistinguishable from that written by humans, mirror the behavior of people in social situations (Manning et al., 2024), and even capture the nuances of human behavior in economic, psycholinguistic, and social psychology experiments (Aher et al., 2023). In the context of democratic processes, LLMs have demonstrated capabilities in a wide range of tasks: inferring disposition of politicians (Wu et al., 2023), lobbying government entities (Nay, 2023), predicting election results in various countries around the world (Bradshaw et al., 2024; Gujral et al., 2024; von der Heyde et al., 2024; Yu et al., 2025), and reporting the opinions of the populace on various policy issues (Sanders et al., 2023; Lee et al., 2024; Argyle et al., 2023).

Current strategies of inducing LLMs to report public opinion on policy issues closely follow traditional polling strategies. LLMs are asked to respond to a series of polling questions, as though they were a person filling out a survey (Sanders et al., 2023; Argyle et al., 2023). In particular, the LLM is instructed to mirror the opinions of an individual belonging to a particular demographic group. By repeatedly querying the LLM with different questions and prompting the LLM to represent different individual personas, the end result is a series of synthetic responses similar to those that a pollster would get through traditional polling methods. Development of such "silicon sampling" techniques (Sun et al., 2024), in conjunction with agentic AI possessing detailed and specific personas, unlocks the possibility of simulating the emanations and dynamics of the human population (Kang et al., 2025; Park et al., 2024). Recognizing that constructing opinion distributions through repeatedly querying each question requires significant time, computational, and monetary expenditures, other strategies utilize raw next-token probabilities from a single generation as an estimator for the distribution of responses that would be expected when the query is repeated (Suh et al., 2025; Dominguez-Olmedo et al., 2024).

However, under both strategies, current LLM capabilities fall short of being able to recapitulate traditional human polls with high fidelity or to accurately reflect the distribution of responses. Under traditional silicon sampling techniques, LLMs exhibit high levels of homogeneity in their responses (Yang et al., 2024), particularly in multiple choice and multi-select questions (Steyvers et al., 2025). This can result in a lack of variation in the distribution of LLM-generated responses compared to the distribution of human responses on the same question (Bisbee et al., 2024; Suh et al., 2025). Other studies have found that when attempting to conduct a distribution reconstruction from next-token generation probabilities, both under-estimation and over-estimation of response homogeneity for the distributions have been observed (Suh et al., 2025; Dominguez-Olmedo et al., 2024). These performance issues are further compounded by systematic biases between demographic groups: LLMs are observed to less accurately represent the views of minority groups (Qu & Wang, 2024), and naturally default to representing the views of individuals with higher education and socioeconomic status (Miotto et al., 2022). A handful of studies have also observed that LLM performance on policy questions vary between questions of differing topics, although such studies only examine differences across a few question types (Lee et al., 2024; Qu & Wang, 2024). Correcting such shortcomings in LLM responses is sometimes possible through careful prompt engineering strategies, coaxing the LLM to give more representative results (Sanders et al., 2023; Chen et al., 2024).

If LLMs can accurately model the distribution of responses for political polling questions, they have the potential to revolutionize the polling industry. Techniques for anticipating the perspectives of hard-to-reach individuals can help mitigate the human non-response problem. They can answer thousands, even millions, of questions. And while humans become irritated or fatigued as surveys become longer, potentially resulting in responses unreflective of their true beliefs, LLMs can maintain consistent response quality even as the number of questions becomes arbitrarily large (Qu & Wang, 2024). In addition, gathering responses from an LLM requires a trivial fraction of the financial resources, time, and labor compared to that required to collect equivalent responses from humans. This can open the benefits of political issue polling to elected representatives, candidates, and jurisdictions that do not have the resources to field polls of human respondents.

In this paper, we perform a comprehensive evaluation of how an LLM's performance on predictions varies between different demographic groups and a wide variety of salient policy topics. We do this by systematically

comparing the output of a frontier LLM, OpenAI's GPT-4o-mini, to a large and representative sample of US public opinion on 84 policy issue questions taken from the Cooperative Election Study. In addition, we propose a novel framework and prompt engineering strategy for eliciting responses from the LLM—directly requesting a distribution of responses as opposed to posing as individuals or inspecting token generation probabilities to artificially construct a distribution of responses—and demonstrate that this method yields lower levels of overconfidence and more accurate simulations of human polling responses. We demonstrate that for this particular prompting methodology, the degree of fidelity for the LLM's generated distribution is predictable, and can be quantitatively approximated based only on a priori knowledge: a demographic category and a question of interest.

## 2 Methodology

In order to explore the extent to which LLMs can reproduce distributions of public opinion across a wide range of political issues, we create a framework that queries an LLM with political issue polling questions for which human responses have already been collected (Figure 1). We then analyze how accurately the LLM predicts human responses across differing question types, respondent demographics, and prompt engineering methodologies. Finally, we evaluate the ability to systematically anticipate LLM performance based solely on information available at query-time.

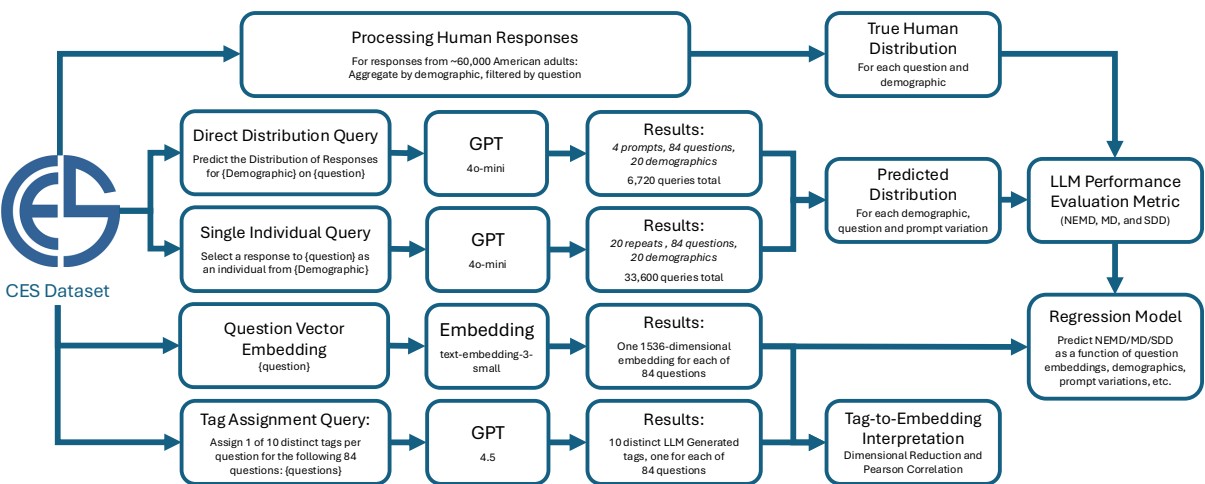

Figure 1: Framework for curating and evaluating LLM predictions on opinion polling questions

### 2.1 Target question and demographic selection

To ensure high-quality human respondent data to judge LLM-generated responses, we used the Final Release of the 2022 Cooperative Election Study (CES) Common Content Dataset (Schaffner et al., 2023).[1] The CES dataset contains a sample of approximately 60,000 American Adults, encompassing a wide range of respondent demographics and relevant policy questions. We focus on the American political environment due to the availability of high quality, large scale, fully granular, publicly available survey data (specifically CES) and because of the outsized role of American opinion polling in US as well as global politics.

From this dataset, we manually selected 84 questions of interest covering a broad range of topics, from healthcare and fiscal policy to social equality and foreign relations. These questions were selected by filtering down to questions centered on public sentiment towards political issues. We removed questions irrelevant to public sentiment, such as questions of factual knowledge (e.g., awareness of candidates running for an office) and personal experience (e.g., whether the respondent had ever run for various political positions). We also

---

[1]While CES datasets are typically released every two years, there was no data release in 2023. The 2022 release was the most recent available at the time this study was conducted. The CES 2024 dataset was released in April 2025 and we will apply our framework to this new data in a future study.

removed questions inherently collinear with the demographics we conditioned on, such as questions about an individual's political leaning. Each selected question was a multiple-choice response on a binary or Likert scale, with two, four, or five distinct response options, referred to as the Cardinality of the question.

We also selected three demographics by which to categorize individuals: Race, Gender, and Ideology. We used a simple categorization for each demographic following Sanders et al. (2023). We categorized Ideology in five bins: Very Liberal, Liberal, Moderate, Conservative, and Very Conservative. We categorized Gender with two bins: Man and Woman. And we categorized Race with two bins: White and Non-white. In total our study thus encompassed twenty distinct demographic permutations for each of 84 questions, yielding a total of 1680 question-demographic permutations.

We acknowledge that sorting the population into merely twenty distinct demographic permutations is a coarse reduction, far from sufficient to encompass all human experience and individuality—especially regarding race and gender. We chose these broad categorizations to enable statistical comparisons with human respondent demographic groups that are well sampled within the CES dataset. Future works should explore the performance of LLM predictions when given more demographic factors and more granular categorizations. Such future work is particularly important to understanding the potential biases and limitations of LLM's ability to recapitulate the perspective of diverse human populations.

## 2.2 Opinion prediction with large language models

To predict the extent to which LLMs can replicate the political opinions of human respondents from the CES dataset, we made use of OpenAI's GPT 4o-mini LLM model, queried using the `langchain` Python library. GPT 4o-mini is a Generative Pre-trained Transformer Language Model with low cost and fast response generation suitable for large scale querying. Due to resource limitations, we focused only on one model in our analysis. We chose instead to focus the resources available for this study on robustly sampling across variations in political issue questions, demographics, and prompting methodologies.

While the specifics of GPT 4o-mini's training data are undisclosed, its training cutoff is October 2023. We recognize this is later than the initial publication of the CES 2022 Dataset, which occurred in March 2023. Although it is unlikely that demographic-grouped summary statistics for CES surveys would be directly in the training data, we acknowledge there is the possibility of data leakage, that the LLM may have been trained on documents containing summary statistics for the CES 2022 survey results. This ambiguity is a persistent challenge in evaluating LLMs with undisclosed training data.

For each of the 1,680 question-demographic permutations, we utilized two different frameworks to elicit a distribution of responses: Single Individual Querying and Direct Distribution Querying. Under the **Single Individual (SI)** querying framework, we implemented the broadly-utilized prompting strategy of instructing the LLM to respond to questions with a single categorical selection, as if the LLM were a member of the demographic group in question, filling out a survey. In addition, we made use of chain-of-thought prompting, requesting the LLM generate a justification before reporting its categorical response. We constructed a distribution of responses by repeating each request twenty times for each question-demographic permutation, for a total of 33,600 distinct LLM queries.

Under the **Direct Distribution (DD)** querying framework, we implemented an uncommon approach, issuing a single request per demographic-question permutation, requesting the LLM directly respond to a survey question with the numerical distribution of responses for the specified demographic. In addition to testing across questions and demographics, we tested the effect of variations in the base query prompt template on the LLM's response. In particular, we evaluated variations in performance when requesting a chain-of-thought justification for the distribution (referred to as the **Chain-of-thought Reminder**) as well as the presence of a reminder that the predicted distribution need not be symmetric, normally distributed, or be non-zero for all Likert categories (referred to as the **Distribution Reminder**). In total, we issued 6,720 queries to the LLM under the DD framework.

We made use of OpenAI's structured JSON response option to ensure that the responses to the SI Querying framework consisted of a single integer and a justification string, and that the responses to the DD Querying

framework was a list of floating-point numbers and an optional justification string. Examples of prompt templates can be found in Appendix A.1.

## 2.3 Quantitatively differentiating questions by topic

To determine the performance of the LLM's prediction across questions of differing topics, we utilized the OpenAI `text-embedding-3-small` model to generate vector embeddings of each question text, creating a quantitative representation of the semantic content for each question. At the same time, to provide interpretable categorizations, we labeled each question with a textual topic tag. We automated and standardized this tagging process by using an LLM: we provided the OpenAI GPT 4.5 model with the 84 questions, requesting that it generate a set of tags that categorized the questions, and to label each question with a single appropriate tag. Given that the tagging process required only a single query, cost was not an important consideration, so we utilized the top performing model from OpenAI available at the time, rather than a lower cost model.

We then interpreted the semantic meaning of the vector embedding space in terms of the questions embedded into it. First, we used a UMAP reduction to two dimensions to visualize relations between embeddings and LLM-assigned tags (Figure 2), finding that questions with the same tag cluster together well. Furthermore, tags with similar semantic meaning (e.g. "Gun Policy" and "Police & Criminal Justice") are also in close proximity. To quantitatively measure the association strength between each embedding dimension and tag, we calculated Pearson correlation coefficients between each binary tag label and the continuous embedding values (on the range $[-1, 1]$) across questions.

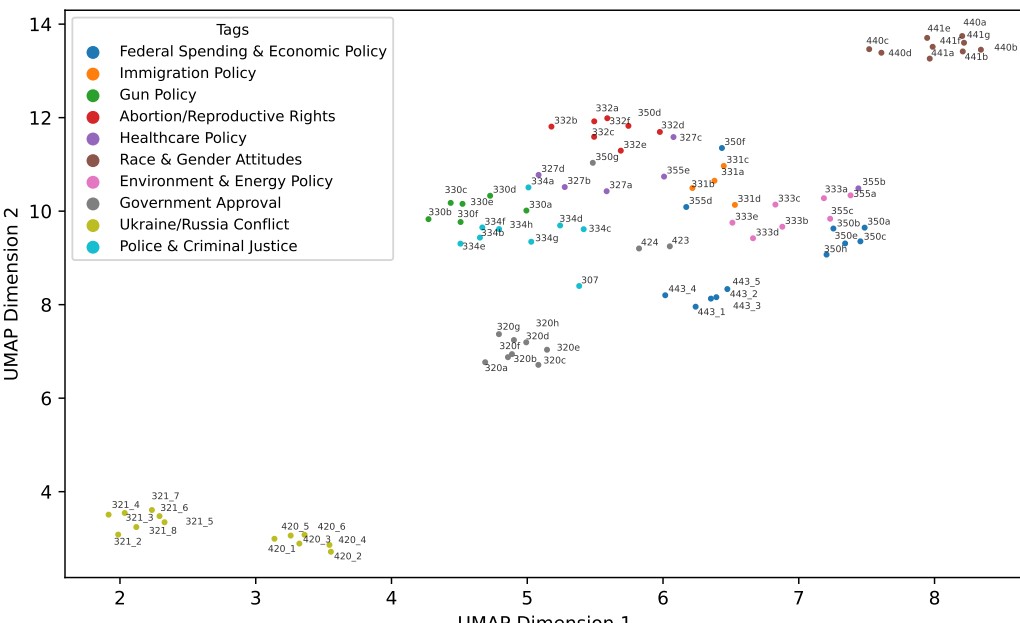

Figure 2: Vector embedding of survey questions mapped to two dimensional space with UMAP and color coded by LLM-assigned content tag.

## 2.4 Processing human responses

To serve as a comparison against the LLM's predicted distributions, we aggregated the individual human responses for each question by the same twenty demographic permutations used to query the LLM. For each question, we dropped individuals who cannot be classified into one of the twenty demographic permutations, primarily those who chose not to respond to demographic identifying questions. Unfortunately, our simplistic demographic categorization also involved dropping individuals with non-binary gender, although this represents

only a small subset of the surveyed population (0.73%). The aggregated individual responses within each demographic permutation then formed the human respondent distributions to be compared against the LLM's predicted distributions.

## 2.5 Comparing human and LLM predicted responses

We compared the distribution of human respondents with the LLM's predicted response distribution for each Question-Demographic permutation using three metrics. **Mean Difference (MD)**, the absolute difference between the mean of the two distributions, measures how much the *average sentiment* of the LLM prediction deviates from the Human distribution. **Standard Deviation Difference (SDD)**, the difference between the standard deviation of the human and LLM distributions, measures the extent to which the LLM overestimates or underestimates the *homogeneity*—or spread—of the Human distribution. **Normalized Earth-Mover's Distance (NEMD)**, also known as the Wasserstein distance, measures the *overall conformity* between the two distributions. Details on each metrics and its interpretations can be found in Appendix A.2.

Our primary interest is quantifying how overall performance (NEMD) varies between different demographics, questions, and prompting strategies. For any statistically significant perturbation in NEMD, the MD and SDD metrics can be used to determine whether this perturbation was due to a biased prediction or an under/over-confident prediction. As such, we fit regression models with each metric individually as a target. In particular, we used a Bayesian Ridge Regressor as implemented in the `sklearn` Python library, examining the performance of the DD querying framework in comparison to the traditional SI querying method, as well as the DD querying framework across the varied querying templates. Details on the regression features, interaction features, and other considerations can be found in the Appendix A.3.

When evaluating regression model performance, we examined the predictive performance of the fitted models on testing data from a stratified train-test split, with data partitioned by question, mimicking the primary use case of the frameworks: to predict response distributions across demographics on question for which human polling results have not been collected.

# 3 Results

We systematically reviewed the performance of the LLM frameworks in reproducing the distribution of human responses, as collected in the CES survey, for each question-demographic permutation, following the configurations described in Section 2.5.

## 3.1 Single Individual and Direct Distribution querying frameworks

We found that the DD framework generally outperformed the SI framework across all our studied metrics.

First, the DD framework performed better in predicting average sentiment (MD) across question-demographic combinations than the SI framework. The DD framework had a lower MD in 1222 out of 1680 distributions (72.7%), exhibiting an MD lower than the SI framework's by 10.4% of the response range[2] on average, with a calculated $2\sigma$ standard error confidence interval of [9.5, 11.3].

Second, the DD framework exhibited better overall conformity to the human distribution (NEMD) than the SI framework in 1299 out of 1680 distributions (77.3%), with an NEMD 0.107 lower than the SI framework on average, with a calculated $2\sigma$ standard error confidence interval of [0.100, 0.113].

Third and finally, the DD method better represented the human respondent's spread in response distribution within each question. When comparing the standard deviations for the Human response distribution with the standard deviation for the distributions generated under the DD and SI frameworks (Figure 5), we found that the DD standard deviations exhibited fairly accurate recapitulations of the human standard deviation. The DD responses successfully reproducing the range of homogeneity observed across human distributions. In contrast, LLMs under the SI framework consistently over-estimated distribution homogeneity

---

[2]Distribution means are scaled to be on the range 0 to 1. As such 10.4% lower on the response range is indicative of a lower difference from the true Human distribution mean by 0.104 on the 0 to 1 scale

(had consistently lower standard deviation). Overall, the DD framework underestimated human respondent homogeneity, with an average standard deviation 0.042 higher than the human standard deviation ($2\sigma$ interval [0.036, 0.048]). On the other hand, the SI framework significantly overestimated distribution homogeneity, with an average distributional standard deviation 0.423 less than the human standard deviation ($2\sigma$ interval [0.408, 0.438]). We note that this large overestimation is due primarily to a high number of distributions for which the SI framework returned identical selections across all queries, resulting in a standard deviation of 0, seen in Figure 4b. This also results in the SI framework outperforming the DD framework NEMD in some specific outlying instances (Figure 3a). When the underlying human distribution is highly homogeneous, the DD framework underestimated the homogeneity of the human distribution, while SI framework produced no variance, overestimating homogeneity, but resulting in a better prediction overall.

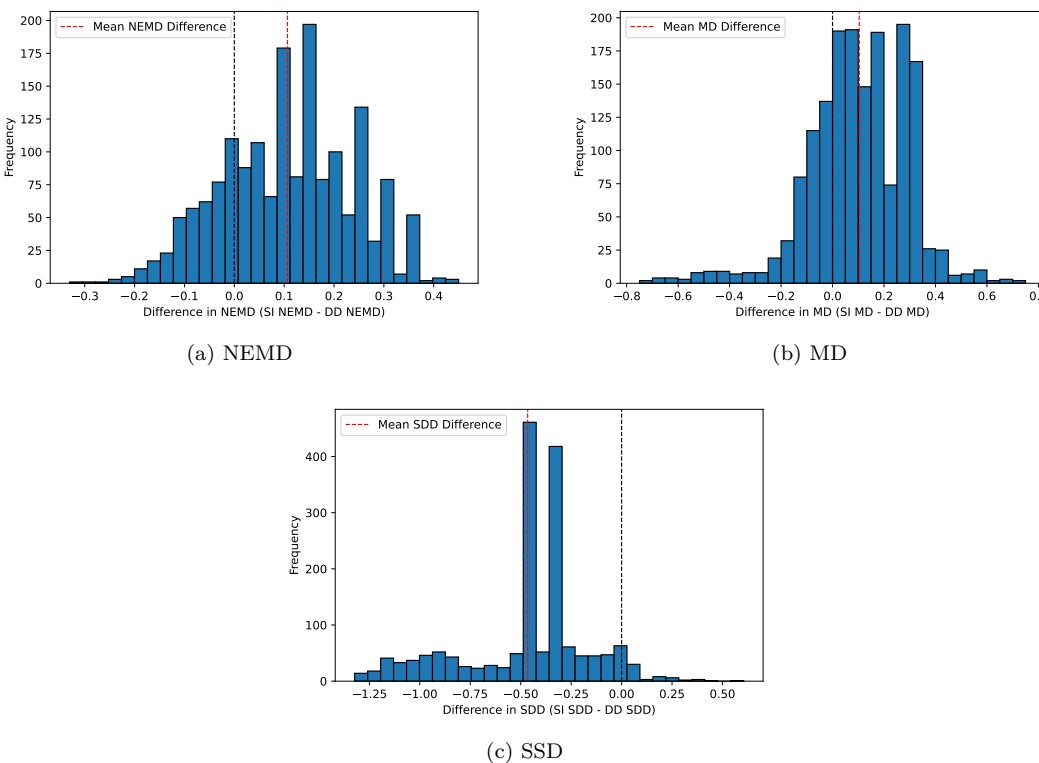

(a) NEMD

(b) MD

(c) SSD

Figure 3: Histograms of pair-wise metric differences between SI and DD Frameworks

## 3.2 Direct Distribution framework performance

To characterize how the DD Framework's performance varies between questions, different demographic combinations, and different prompting strategies, we train a Bayesian Ridge Regression model to predict the NEMD, MD and SDD between the human and LLM response distributions. We begin by training a "naive" variation of the regression model without any interaction terms, to establish a baseline with simple interpretability of coefficients.

Examining the coefficients of the naive model, we found that no individual feature had an outsized impact on predicting the NEMD performance of the LLM. As seen in Figure 6a, many features had relatively small but statistically significant impacts on NEMD. We note that although the absolute magnitude of the coefficients for statistically significant one-hot encoded features may be small, the coefficients can be seen to be more substantial when compared relative to the mean and standard deviation of the metric values. In particular, the statistically significant coefficients were typically on the order of 10 to 20% the metric's distributional

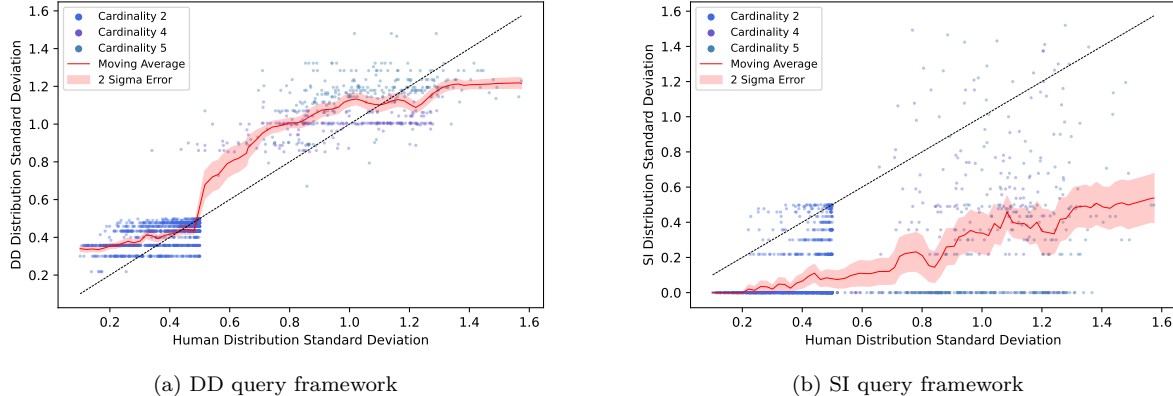

(a) DD query framework  (b) SI query framework

Figure 4: LLM response heterogeneity (standard deviation) for the DD and SI frameworks versus the heterogeneity of human responses. Moving average and associated $2\sigma$ Standard Error shown

spread, as measured by standard deviation, with some being greater than 50% the standard deviation of the metric (Table 1).

### 3.2.1 Demographic and query related coefficients

Table 1: Bayesian Ridge Regression Coefficients for one-hot encoded features across each metric for the DD framework

| One-Hot Encoded Features | Metrics[*] | | |
|---|---|---|---|
| | NEMD | MD | SDD |
| Ideology: Very conservative | **0.016** | **0.065** | **-0.016** |
| Ideology: Conservative | **0.015** | **0.046** | **0.018** |
| Ideology: Liberal | 0.003 | **0.008** | **0.032** |
| Ideology: Very liberal | -0.002 | **0.013** | **0.027** |
| Race: non-white | -0.001 | -0.004 | **-0.019** |
| Gender: Woman | 0.001 | -0.001 | 0.001 |
| Prompt: Chain of Thought | **-0.003** | **0.007** | **-0.013** |
| Prompt: Distribution Reminder | 0.001 | **0.009** | -0.003 |
| Cardinality: 2 | -0.002 | -0.001 | -0.024 |
| Cardinality: 4 | -0.008 | -0.003 | 0.008 |
| Metric Mean | 0.090 | 0.147 | 0.046 |
| Metric Standard Deviation | 0.068 | 0.140 | 0.124 |
| Metric Minimum | 0.000 | 0.000 | -0.420 |
| Metric Maximum | 0.396 | 0.846 | 0.591 |
| Bayesian Ridge Model $R^2$ | 0.289 | 0.291 | 0.385 |

[*] Statistically significant coefficients at the $p = 0.05$ level bolded

Examining overall conformity to human distributions (NEMD), we found that coefficient values for race and gender features tended to be small relative to metric spread (standard deviation), and—with the exception of the SDD metric for race—not statistically significant, indicating that after controlling for question variations via embedding features, LLMs predict distributions with roughly equivalent fidelity across racial and gender demographics. On the other hand, ideological features—particularly on the conservative end—tend to have the largest significant coefficients, suggesting that the LLM may struggle to correctly recapitulate the issue preferences of conservative demographic groups.

Examining the MD metric for Ideology coefficients, we found that all four coefficients were positive and statistically significant. This indicates that the MD metric—measuring how much the average sentiment (mean) deviates between the LLM and Human distributions—increases as one moves to more extreme ideological groups. In particular, the model is worse at predicting average sentiment when one moves right-of-center or left-of-center on the political spectrum, with the affect more pronounced in conservative ideological demographics. As for the SDD metric, we found that all ideology coefficients are statistically significant. In particular, given that the average SDD was positive, LLMs under the DD framework most severely underestimated distribution homogeneity for the "Liberal" ideological group, followed by the "Very Liberal" ideological group, and least severely underestimated distribution homogeneity of the "Very Conservative" ideological group.

Examining prompt variations, we found that the addition of the Distribution Reminder only had a statistically significant impact on the MD metric, and not on the NEMD or SDD metrics. In particular, the coefficient for the Distribution Reminder feature is positive, indicating that the presence of a Distribution Reminder in fact increased the difference in the average sentiment predicted by the LLM distribution and that of the true Human distribution. Although we note that relative to the standard deviation of the MD metric (0.140), this coefficient is relatively small, on the magnitude of approximately 6% of the metric spread. This indicates that the Distribution Reminder induced a marginal, but statistically significant, decrease in MD performance and no significant impact on the other metrics.

For the Chain-of-Thought Reminder, we found a small but statistically significant impact on all three metrics. In particular, the coefficient for NEMD is negative, positive for MD, and negative for SDD. Thus, the presence of a Chain of Thought reminder improved the overall similarity between the LLM and Human distributions (as measured by NEMD) by decreasing the extent to which the LLM underestimates the homogeneity of distribution (as measured by SDD), at the cost of a marginally larger deviation in average sentiment (as measured by MD). We note that relative to the standard deviations of the three metrics, the affect of the Chain-of-Thought Reminder is only marginal, but nonetheless statistically significant, representing only 4%, 5%, and 10% of the metric spread for NEMD, MD and SDD respectively.

Finally, we note that neither cardinality feature yielded statistically significant impacts on any of the three metrics. This result is of particular importance to the SDD metric, wherein this establishes that variance in SDD—or the extent to which the model overestimates or underestimates the homogeneity of the true Human distribution—is primarily explained by features of the query (such as demographics, query prompt reminders, and embeddings representative of the question types) as opposed to the number of option bins a particular question has.

### 3.2.2 Vector embedding feature coefficients

Interpreting question embedding features enables an understanding of how varying topics influence the predictive ability of the DD framework. Thus, we calculated Pearson Coefficients for each tag-embedding dimension combination. Table 3 contains calculated corelation coefficients for each tag and select embedding features identified as statistically significant by the Bayesian Ridge model.

By cross-referencing corelation coefficients with the regression model coefficients, we can deduce the effect that variation across topics (tags) have on performance. For instance, Feature 87 had the largest magnitude coefficient for predicting NEMD (Figure 6a), and the coefficient was negative. Referring to Table 3, we found a strong positive correlation between Feature 87 and questions assigned the tag "Abortion/Reproductive Rights." As such, we postulate that the LLM performed better in overall conformity (NEMD) on questions relating to "Abortion/Reproductive Rights." Similarly, examining the tag "Race & Gender Attitudes," we found a strong positive correlation between the tag and Feature 35, which itself had a negative coefficient in the Bayesian Regression model. As such, we also postulate that questions with this tag tend to be predicted better (lower NEMD) by the LLM than the average question.

We note that while it is possible—and more conducive to model interpretability—to utilize tags directly as features for the regression models, we found reduced predictive performance due to the lower granularity differentiation between questions of varying topics.

### 3.3 Predicting performance per question

The simple regression model presented in Section 3.2 suggests the possibility of using a predictive model to anticipate LLM performance on recapitulating the distribution of public opinion as a function of differing question types and respondent demographics.

We see in previous evaluations that there exist features with statistically significant predictive power for each of the three metrics, even under a naive Bayesian Ridge model without interaction terms. As such, there exists promise in the use of the Bayesian Ridge model in predicting the performance of the LLM *a priori*, using only information available from the query: the demographics, the presence/absence of specific reminders, and the embedding vector of the question text itself.

Table 2: Predictive abilities of various regression models for each metric as evaluated through in-sample (training) and out-of-sample (testing) $R^2$ under the DD querying framework

| Regression Model | NEMD | | MD | | SDD | |
|---|---|---|---|---|---|---|
| | Training | Testing | Training | Testing | Training | Testing |
| Bayesian Ridge | 0.287 | 0.312 | 0.328 | 0.175 | 0.394 | 0.378 |
| Bayesian with Interactions | 0.474 | 0.526 | 0.473 | 0.268 | 0.732 | 0.687 |
| Gradient Boosting | 0.415 | 0.517 | 0.442 | 0.273 | 0.598 | 0.655 |

We found that, as seen in Table 2, when evaluated on testing data, the Bayesian Ridge Model with interaction terms has an $R^2$ of 0.526, 0.268, and 0.687 when predicting NEMD, MD, and SDD respectively, indicating that the Bayesian Ridge model with interaction terms has strong capabilities in predicting the performance of the LLM model, based only on information available before the query is made. We found that the Bayesian Ridge model with interactions achieved similar testing performance to the gradient boosting model, which was selected as a baseline evaluation for performance. Thus, given that the Bayesian Ridge model with interactions offers comparable predictive performance alongside greater interpretability, we focused primarily on the Bayesian Ridge model.

Such predictive models enable users of AI polling tools to anticipate whether an LLM is likely to recapitulate the human distribution well (NEMD prediction), but also whether that recapitulation is likely to exhibit a biased average sentiment (MD prediction), and the extent to which the recapitulation is likely to reflect the homogeneity of the true Human distribution (SDD prediction).

Finally, it should be noted that employing similar strategies for the SI framework proved ineffective. As seen in Table 4, the out-of-sample $R^2$ values for both Gradient Boosting and Bayesian Ridge models in predicting NEMD, MD, and SDD for the SI method were consistently negative, indicating that the population mean was a better predictor than the fitted model. As such, while the predictive regression method is applicable for DD framework performance, it is not suitable for the SI framework. While we only test a limited range of feature engineering and predictive model choices here, this finding suggests that the DD methodology may generally be more predictable in its performance than the SD.

## 4 Discussions

We find that directly querying LLMs for the response distribution of a political issue within a demographic group (the DD framework) generally offers higher predictive performance than simulating individual, independent responses (the SI framework). The DD method is superior in its ability to reproduce the mean response of the humans in each demographic cell (MD metric), superior in overall conformity to the response distribution (NEMD metric), and—most of all—superior in its ability to reproduce the actual spread in human responses to the question (SD metric).

For independent SI queries, the LLM is strongly biased towards returning the most probable response, often resulting in little to no variance in the distribution of SI queries. However, if one suspects a high degree of homogeneity inherent in the human response distribution, this shortcoming of the SI framework can

become a feature; in this regime, DD may underestimate the response homogeneity. In addition, it should be acknowledged that the SI framework—although less performant in predicting the distribution of human opinions—enables a different polling paradigm with sequential follow-up questions, simulating an individual explaining their thought process for a specific response or opinion, progressively updating their response following interventions such as exposure to a political message or new factual information.

Broadly, we see that the DD framework's performance varies with some predictability across different demographic groups. Particularly, across ideological groups, the LLM appears to predict the average sentiment (as measured by MD) of the ideologically "Moderate" populace with the greatest fidelity, decreasing as the Ideology becomes more extreme in either direction of the political spectrum (increasing MD). In addition, LLMs under the DD framework appear to consistently underestimate the homogeneity of the true human distribution across ideological groups, with this effect being the most pronounced for the "Liberal" ideological group, and least pronounced for the "Very Conservative" ideological group.

A similar phenomena of varied underestimation is seen across the two studied racial demographic groupings, with the LLM more severely underestimating the homogeneity of the human distribution for the "White" racial group. Finally, the LLM appears to predict the sentiment of the "Man" and "Woman" demographic groups with roughly equal levels of fidelity, with the results of neither populace being statistically different from the other for any of the three metrics.

Finally, the DD framework performance across all metrics exhibits predictability based only upon information taken from the question and target demographic. As such, it is possible to assign confidence measures to how well we expect any particular LLM prediction to be under this framework. For many applications of AI polling, this may be a decisive advantage: the uncertainty in the simulated polling response on the SI method is essentially unbounded, while the uncertainty in the DD method is at least somewhat predictble.

Future work should extend upon the capabilities of LLMs characterized here to advance practical AI tools for use in social science and applied contexts. Given that the most valuable use of LLMs in political polling contexts will likely be *augmenting*, as opposed to fully *replacing*, human surveys and panels (Wang et al., 2024), more work is needed to understand how these models will perform in boosting the accuracy of population estimates from finite sample sizes. Second, given the robust ability to predict the issue stances of discrete demographic groups, population-level estimates across demographics can be achieved (and their performance characterized) using techniques like post-stratification (Cerina & Duch, 2023).

This study has several limitations. Evaluating LLM framework performance against only the CES dataset does not address epistemic uncertainties on the degree to which GPT 4o-mini broadly reflects the characteristics of AI models or on the degree to which the CES dataset itself conveys ground truth human belief. While the CES dataset offers one of the most comprehensive political surveys across a wide range of political issues and with sample size sufficient to measure robust demographic variation, future work should evaluate additional survey datasets and LLMs. In addition, this study utilizes broad demographic bins, and is limited to topics covered across 84 questions; examining the performance of the frameworks under higher granularity demographic bins and across a wider variety of questions would also be of great importance in further evaluating the efficacy of LLMs for the purpose of political polling. Furthermore, examining further variations in the LLM querying strategy, such as temperature and other prompting strategies, should be explored in greater depth.

## 5   Conclusion

In comparison to traditional polling techniques, LLMs enable an estimation of public opinion on a variety of political issues at far lower cost. In particular, the proposed Direct Distribution framework for eliciting LLM predictions on public opinion exhibits stronger performance than previously proposed frameworks that query from the perspective of a single individual. Furthermore, the proposed framework exhibits predictable variability across differing questions and demographics, creating the possibility of an AI-augmented political polling paradigm which utilizes the LLM moreso when predicted distributions are likely to be high fidelity, and direct polling of human respondents moreso when the LLM's anticipated performance is low.

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

# A Appendix

## A.1 Prompt templates utilized in the DD and SI frameworks

### A.1.1 SI framework query template

Please represent the opinion of an individual selected at random from the politically {Ideology} {Gender} populace living in the United States who identify as {Race} on the policy proposal to: "{Question}" where the person's position on the issue is on a {Cardinality}-point scale, with 1 representing the position "{low_level}" and {Cardinality} represents the position "high_level". Write a brief 1 sentence justification on the beliefs of the selected {Race} {Ideology} {Gender} individual which motivates the score. Following the justification, report the score the individual would select. Report the justification and score via the "justification" and "score" JSON keys.

### A.1.2 DD framework query template with both chain-of-thought and distribution reminders

Please represent the distribution of opinions of the politically {Ideology} {Gender} populace living in the United States who identify as {Race} on the policy proposal to: "{Question}" where a person's position on the issue is on a {Cardinality}-point scale, with 1 representing the position "{low_level}" and {Cardinality} represents the position "high_level". Write a brief 1 sentence justification on the beliefs of the selected {Race} {Ideology} {Gender} populace, and infer the mean and spread of the distribution. Note the distribution need not be normal, symmetric, or encompass all category options. Following the justification, report the proportion of individuals that would select each position as a list of decimals, such that the sum of all decimals is 100. The list should contain exactly {Cardinality} numbers. Report the justification and distribution via the "justification" and "distribution" JSON keys.

### A.1.3 DD framework query template with no chain-of-thought or distribution reminders

Please represent the distribution of opinions of the politically {Ideology} {Gender} populace living in the United States who identify as {Race} on the policy proposal to: "{Question}" where a person's position on the issue is on a {Cardinality}-point scale, with 1 representing the position "{low_level}" and {Cardinality} represents the position "high_level". Report the proportion of individuals that would select each position as a list of decimals, such that the sum of all decimals is 100. The list should contain exactly {cardinality} numbers. Report the distribution via the "distribution" JSON key. Leave the "justification" JSON key as an empty string: Do not report any justification for the distribution.[3]

---

[3]Note that we also utilized a DD framework query template with the Chain-of-thought Reminder but no Distribution Reminder, as well as a query template with the Distribution Reminder but no Chain-of-thought Reminder. However, given these templates follow the exact same structure as the two featured DD framework query templates, and are derived by combining sentences present in the featured templates, they are removed to reduce redundancy

## A.2 Details on metric calculation and interpretations

**Mean Difference (MD)** is the absolute difference between the mean of the Human respondents' distribution and of the LLM's predicted distribution. We standardized the mean based on the cardinality of the question such that the mean of each distribution is in the range 0 to 1. Given that the mean of the distribution conveys the *average sentiment* across a demographic sub-population, the MD metric offers a measure as how much the LLM's predicted average sentiment deviates from the true average sentiment of the human respondents, for any particular question-demographic combination.

**Standard Deviation Difference (SDD)** is the difference between the standard deviation of the Human respondents' distribution and of the LLM's predicted distribution. Given that the standard deviation measures the *homogeneity*—or the spread—of the distribution, the SDD metric offers a measure as to how well the homogeneity of the LLM's predicted distribution mirrors the homogeneity of the true human distribution for any particular question-demographic combination. In particular, a negative SDD indicates that the LLM *overestimates* the homogeneity of the human distribution, and conversely, a positive SDD indicates that the LLM *underestimates* the homogeneity of the human distribution.

**Normalized Earth-Mover's Distance (NEMD)**—also known as the Wasserstein distance—between the Human respondents' distribution and the LLM's predicted distribution is also computed. The NEMD is useful for quantifying the *overall conformity* of the LLM's predicted distribution to the true Human distribution, accounting for both the conformity in average sentiment and homogeneity, making it a useful indicator of overall performance. Specifically, a low NEMD implies both a low MD and a low SDD. On the other hand, the MD and SDD metrics are useful in interpreting the cause for a high NEMD; one can determine whether the high NEMD is caused by a biased LLM prediction that fails to recapitulate *average sentiment* across a demographic group (resulting in a larger MD), or by *over-estimating or under-estimating the homogeneity* of the distribution, (resulting in an SDD farther from 0). We further note that for questions with a response cardinality of 2 (binary response questions), the SDD and MD will be highly correlated and monotonically related, making SDD primarily of interest for higher cardinality questions.

## A.3 Details on regression model features

To examine the performance of the DD querying framework in comparison to the traditional SI querying method, we examined each metric across both of the querying frameworks, evaluating whether or not there exist systematic differences in performance between the two methods for specific question types or demographics, by calculating the pair-wise difference for each metric across all 1680 Question-Demographic permutations, training a regression model to predict this difference value as a function of the demographic permutation and of the question embeddings.[4] We also examined SDD metric values to interpret whether differences in NEMD value were due to under-confidence or overconfidence in each of the querying frameworks. Finally, when modeling the performance of the DD querying framework, we included two binary features representing the base prompt template used: in particular, whether the template included the Chain-of-Thought reminder, and whether the template included the Distribution Reminder. The repetition of the DD querying framework across prompt variations also serves as an internal consistency check on the reproducibility of the predicted distribution generated by the LLM.

In each regression model, we represented each demographic category as a separate one-hot encoded feature. This resulted in six features, one for each of Race and Gender, as well as four for Ideology, such that the reference categories are "White," "Man," and "Moderate" for Race, Gender and Ideology respectively. We also controlled for the cardinality (number of likert-scale response options) of the question via two one-hot encodings for cardinality values of two and four, with the reference category being cardinality of five. In addition, we represented each dimension of the vector embedding as a separate feature. Following the OpenAI embeddings model documentation,[5] we made use of Matryoshka representation learning (Kusupati et al.,

---

[4]For the purposes of comparing against the SI framework, we do not use results from all four prompting techniques utilized in the DD framework. We used the DD query results generated from the prompt template containing the Chain-of-thought Reminder, but not containing the Distribution Reminder, which is the closest matching template to the one used in the SI querying framework.

[5]https://openai.com/index/new-embedding-models-and-api-updates/

2022) to shorten the question embeddings to size 100 to improve the interpretability of the regression model. We normalized each of the embedding dimension features via the `sklearn.preprocessing.StandardScaler` class, saving the fitted StandardScaler instance (such that a StandardScaler fit to training data embedding features may be used to normalize testing data embedding features, for instance). We also implemented variations of the regression models with interaction terms between each of the six one-hot encoded demographic features and and all 100 embedding dimension features.

### A.4 Supplemental results and figures for tags and vector embeddings

Table 3: Pearson correlation between each tag and statistically significant embedding features (as determined through the bayesian ridge model)

|  | Feature Number[*] | | | | |
| Tag | 87 | 14 | 97 | 38 | 35 |
| Abortion/Reproductive Rights | **0.43** | -0.01 | 0.06 | 0.07 | -0.16 |
| Environment & Energy Policy | -0.03 | -0.20 | -0.11 | -0.22 | -0.06 |
| Federal Spending & Economic Policy | -0.18 | 0.22 | **-0.40** | **0.30** | 0.03 |
| Government Approval | 0.08 | 0.18 | **0.33** | 0.15 | 0.10 |
| Gun Policy | 0.14 | 0.17 | 0.01 | -0.04 | 0.16 |
| Healthcare Policy | -0.14 | -0.12 | -0.08 | 0.05 | -0.20 |
| Immigration Policy | -0.04 | -0.02 | 0.21 | -0.14 | -0.16 |
| Police & Criminal Justice | -0.17 | **0.39** | -0.10 | 0.06 | -0.03 |
| Race & Gender Attitudes | -0.03 | -0.13 | -0.02 | 0.04 | **0.48** |
| Ukraine/Russia Conflict | -0.01 | **-0.46** | 0.14 | **-0.34** | -0.21 |

[*] Pearson Coefficients with magnitude greater than 0.25 bolded

### A.5 Predictive models for the SI framework

Table 4: Predictive abilities of various regression models for each metric as evaluated through in-sample (training) and out-of-sample (testing) $R^2$ under the SI query framework

| Regression Model | NEMD | | MD | | SDD | |
| | Training | Testing | Training | Testing | Training | Testing |
| Bayesian Ridge | -0.310 | -0.156 | -0.163 | -0.296 | 0.126 | -0.158 |
| Bayesian with Interactions | -0.559 | -0.091 | -0.443 | -0.308 | 0.079 | -0.213 |
| Gradient Booster | -0.029 | -0.103 | -0.036 | -0.448 | -0.114 | -0.441 |

Table 5: Predictive abilities of various regression models for metric differences (SI - DD) between frameworks, as evaluated through in-sample (training) and out-of-sample (testing) $R^2$

| Regression Model | NEMD | | MD | | SDD | |
| | Training | Testing | Training | Testing | Training | Testing |
| Bayesian Ridge | -0.363 | -0.106 | -0.336 | -0.136 | 0.071 | -0.131 |
| Bayesian with Interactions | -0.587 | -0.103 | -0.477 | -0.153 | 0.033 | -0.175 |
| Gradient Booster | -0.003 | -0.187 | 0.026 | -0.217 | -0.095 | -0.627 |

Fitting various regression models to predict metric differences between the SI and DD frameworks, we find that the models have negative out-of-sample $R^2$ values, as seen in Table 5, indicating that the models perform

worse than simply using the average metric difference. As such, we conclude that there does not exist a systematically predictable performance difference between the SI and DD frameworks across demographic groups or question types (as represented through embedding dimensions as model features). This result is due primarily to the fact that the results of the SI framework are inherently not predictable, while the results of the DD framework are, resulting in the difference between the frameworks across metrics being unpredictable as well.

### A.6 Supplemental results for the NEMD, MD and SDD metrics

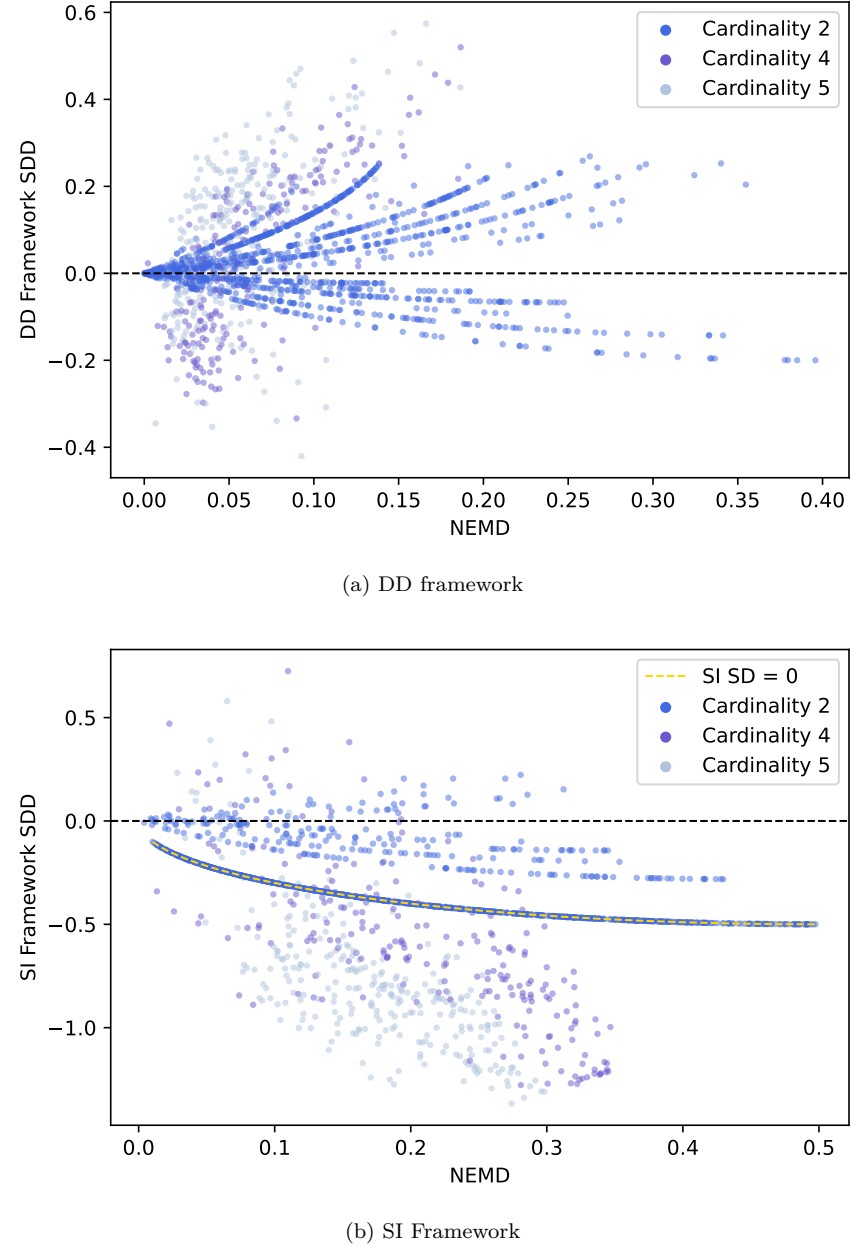

(a) DD framework

(b) SI Framework

Figure 5: DD and SI framework SDD metric as a function of NEMD, with instances in which the LLM returned the same response across all trials (Standard Deviation of 0) for SI framework identified.

### A.7 Supplemental results for the Bayesian Ridge regression models

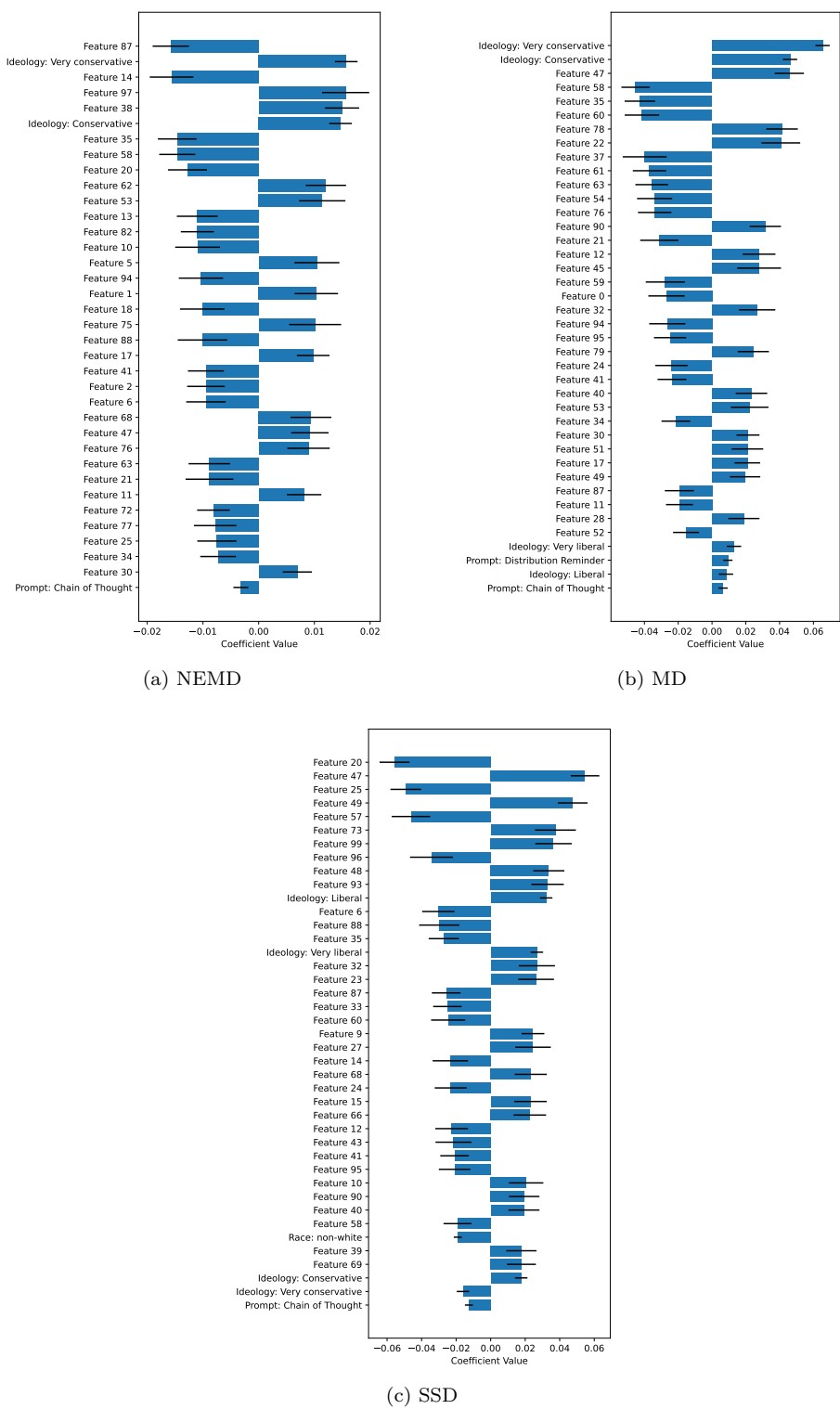

(a) NEMD

(b) MD

(c) SSD

Figure 6: Statistically Significant feature coefficients (p = 0.05) for Bayesian Ridge regression models predicting each metric

Table 6: Bayesian Ridge Regression Coefficients for embedding features across each metric, filtered to Embedding features with statistically significant impact on NEMD

| Embedding Features | Metrics[*] | | |
|---|---|---|---|
| | NEMD | MD | SDD |
| Feature 1 | **0.010** | -0.005 | 0.019 |
| Feature 2 | **-0.009** | 0.005 | -0.010 |
| Feature 5 | **0.010** | 0.005 | -0.001 |
| Feature 6 | **-0.009** | -0.004 | **-0.030** |
| Feature 10 | **-0.011** | -0.001 | **0.021** |
| Feature 11 | **0.008** | **-0.019** | 0.013 |
| Feature 13 | **-0.011** | -0.010 | 0.005 |
| Feature 14 | **-0.016** | -0.020 | **-0.023** |
| Feature 17 | **0.010** | **0.021** | -0.004 |
| Feature 18 | **-0.010** | -0.014 | -0.013 |
| Feature 20 | **-0.013** | -0.009 | **-0.056** |
| Feature 21 | **-0.009** | **-0.031** | 0.002 |
| Feature 25 | **-0.008** | 0.001 | **-0.049** |
| Feature 30 | **0.007** | **0.021** | 0.002 |
| Feature 34 | **-0.007** | **-0.021** | -0.001 |
| Feature 35 | **-0.015** | **-0.043** | **-0.027** |
| Feature 38 | **0.015** | 0.013 | 0.012 |
| Feature 41 | **-0.009** | **-0.024** | **-0.021** |
| Feature 47 | **0.009** | **0.046** | **0.055** |
| Feature 53 | **0.011** | **0.022** | 0.009 |
| Feature 58 | **-0.015** | **-0.045** | **-0.019** |
| Feature 62 | **0.012** | 0.003 | 0.017 |
| Feature 63 | **-0.009** | **-0.036** | -0.003 |
| Feature 68 | **0.009** | -0.002 | **0.023** |
| Feature 72 | **-0.008** | -0.003 | 0.000 |
| Feature 75 | **0.010** | -0.001 | -0.010 |
| Feature 76 | **0.009** | **-0.034** | 0.015 |
| Feature 77 | **-0.008** | -0.004 | -0.018 |
| Feature 82 | **-0.011** | 0.003 | -0.007 |
| Feature 87 | **-0.016** | **-0.019** | **-0.026** |
| Feature 88 | **-0.010** | -0.021 | **-0.030** |
| Feature 94 | **-0.010** | **-0.026** | -0.019 |
| Feature 97 | **0.016** | 0.009 | 0.011 |
| Metric Mean | 0.090 | 0.147 | 0.046 |
| Bayesian Ridge Model $R^2$ | 0.289 | 0.291 | 0.385 |

[*] Statistically significant coefficients at the $p = 0.05$ level bolded

