# OpenReview forum: "Characterizing the ability of LLMs to recapitulate Americans’ distributional responses to public opinion polling questions across political issues"
_TMLR — Rejected by TMLR_

### Review · Reviewer_EctA · 2025-12-16

**Summary Of Contributions:**

This paper looks at using LLMs to generate predictions of political polling. Specifically if LLMs can generate accurate polling prediction to 84 questions of interest based on three factors, gender, race, and political alignment, binning these quantities into 20 bins and not considering minor groups. The paper considers two method of generating said predictions Single Individual (SI) and Direct Distribution (DD). The paper then looks at some analysis of where these methods perform better and worse and using linear models to see which factors are predictive of success.

**Additional Comments:**

Personally I would judge the paper is more in the field of statistics or data science than machine learning.

**Audience:**

Yes

**Audience Explanation:**

Using LLMs to aid political polling, or more generally as a tool for social science is a new field which I'm sure some of TMLR's audience would find interesting.

**Broader Impact Concerns:**

Sadly, I do not feel well place to comment on the Broader Impact of this work, as I have very little knowledge in the area of political polling which is the focus of this work.

**Claims And Evidence:**

No

**Claims Explanation:**

Some important details, formalism, and results are missing. There are some strange choices in methodology. Some conclusions seem over stated given the single dataset considered.

Going into greater detail this paper asks the question how well LLMs can do accurate polling prediction. The results however really only compare DD against SI, results comparing against the ground truth (giving an absolute idea of how well either method performs for polling prediction) seem to be missing.

There also a lot of details of specific constants that seem to be missing (such as for the data set and splits) and hyperparameters missing too, temperature for LLM, priors for Bayesian Ridge Regression etc.

There are some questionable choices in the methodology like using a LLM to cluster 84 questions without any mention a manual check. There is very little formalism presented which make some of the details a little hard to follow especially for the linear models in the second half and what they are used to predict. I also find the section talking about how specific embedding features effect a prediction when these have no real world meaning, confusing and not sure what it adds to the paper.

I think some of the claims such as "the proposed framework exhibiting predictable variability across differing questions and demographics, creating the possibility of an AI-augmented political polling paradigm" is likely overstated given the single data set considered. Important details on the methodology backing up this claim seem to be missing to me. Specifically, detail of the test and train splits for the linear models used.

I think with some slight adjustment the authors could give accurate, convincing and clear evidence. But to me, this paper has not reach this threshold yet.

**Requested Changes:**

- Figure 3 would be better shown as two sets of 3 histograms comparing both methods against the ground truth.
- Some important details are missing: Size of CES data set, and subsets used, size of test train splits, temperatures used when sampling from models,etc (plus descriptions of how were these values selected) .
- It be nice to see the exact formulae for the metrics used.
- Please manually check the clustering of the 84 questions, or provide these clusters to the reader.
- Please avoid using such very blue's in figure 4.
- I'd personally like to see some formalism for the Bayesian Ridge Regression, or adding some more detail here, as I find this section a little hard to follow.

---

> ### Author Response · Authors · 2026-01-11
> **Response to Review**
>
> Thank you for taking the time to review our submission and provide thoughtful feedback.
>
> Comparison against the ground truth: We want to draw the reviewer’s attention to the extensive comparisons between our LLM-based predictions (SI and DD methods) and the ground truth data (from the human CES survey data) already present in the manuscript. Indeed, these comparisons are the focus of this work. All evaluations for the two polling methods are performed using metrics based on direct comparisons between the LLM predictions and ground truth data as measured by the normalized Earth-mover distance (NEMD) statistic, also known as the Wasserstein metric, which is a measure of the difference between the LLM predicted distribution and the corresponding human distribution sourced from one of the largest and most comprehensive datasets of political opinion in the US. As such, the study does indeed provide both an indication of how different methods compare to each other, as well as to the human ground-truth.
>
> Methodological details, training-test splits, and hyperparameters: While omitted from the submitted draft to maintain author anonymity, all code utilized for the entire evaluation framework can be found in a public github repository, which we would link to the final camera-ready manuscript. The code contains seeds for all splits, hyperparameters for all fitted models, as well as the responses collected from querying LLMs and embedding tools. The study will thus be fully reproducible, with all data, model details, configurations parameters, and code necessary to repeat the study publicly available. We will also add the suggested details (temperature, data size, split sizes) as footnotes to the manuscript for additional clarity.
>
> Embedding features and clustering: We agree with the reviewer that it is important to interrogate the features of predictive models, which is why we have included discussion and figures in the paper providing interpretations of the vector embeddings we use. While embedding features are not directly interpretable, as they are vector representations of topics, they are useful features for the predictive models discussed in the study to quantify different question topics. The embedding-based clustering of the 84 questions, color coding, and categorical tags (displayed in figure 2) is included to visualize the relation between the numerical embeddings and the human-understandable topics of the 84 questions, reinforcing the suitability of the embeddings as a quantitative method for enabling predictive models to evaluate performance as a function of question topic. As a note, the assigned clusterings for the 84 questions are available within the repository for readers to view, supporting both reproducibility and interpretability.
>
> Additional figure formatting and coloration changes: we appreciate the advice given in terms of figure layout and coloring and will revise accordingly in preparation for a camera-ready manuscript

---

### Review · Reviewer_U7Ft · 2025-12-18

**Summary Of Contributions:**

This paper investigates whether a large language model (LLM) can reproduce distributional response patterns from public opinion polling questions across a range of political issues and demographic subgroups. This paper compares two elicitation paradigms, single individual (SI) prompting and direct distribution (DD) prompting, and evaluates how closely the resulting outputs align with actual polling responses using various evaluation metrics. This paper reports that DD more closely matches the polling response distributions than SI.

**Additional Comments:**

N/A

**Audience:**

No

**Audience Explanation:**

The topic of using LLMs to approximate polling response distributions is potentially interesting, but the current study does not provide sufficiently reliable or generalizable evidence. And the methodological contribution is limited, as the both approaches are natural baselines. More importantly, the experimental results are hard to interpret due to unresolved validity concerns such as potential data leakage and a narrow evaluation scope across models and datasets. In addition, it remains unclear whether using an LLM is an appropriate approach for measuring future public opinion, given that public opinion can shift quickly and may not be well captured by a model’s fixed knowledge.

**Claims And Evidence:**

No

**Claims Explanation:**

Overall, the evidence presented is not yet sufficiently convincing, or clearly supported, and it remains difficult to assess whether the experimental setup is fully methodologically sound.

1. **Possibility of data leakage.** As the authors also note, the CES 2022 dataset may have been included in the training data of GPT 4o-mini. If so, the reported performance could be influenced by memorization or training data contamination, meaning the results may not reflect the model’s true ability to reproduce polling response distributions. This concern weakens the interpretability of essentially all subsequent findings.

2. **Limited coverage of models and datasets.** Although it is plausible that direct distribution (DD) prompting could outperform single individual (SI) one, the paper evaluates this hypothesis in an overly narrow setting with too few LLMs and too few datasets. Validation across multiple LLMs and additional polling datasets would be needed to establish robustness and to support the paper’s broader claims.

3. **Lack of experiments with RAG-based LLMs.** The introduction discusses model-based listening approaches, but the empirical study does not consider retrieval-augmented generation (RAG) settings that could better approximate such approaches by incorporating up-to-date, external information. Evaluating a RAG-based variant could lead to more informative findings about when and how LLMs can track shifting public opinion. In its current form, the paper relies on a closed, non-retrieval LLM setup, which is inherently limited to extracting distributions from the model’s internal, static knowledge and is therefore unlikely to reflect evolving opinion dynamics in realistic scenarios.

**Requested Changes:**

It would be helpful if the authors could address the three issues raised above. In addition, I would like to see a clearer discussion of whether obtaining future public opinion estimates from an LLM is meaningfully justified, and under what assumptions or use cases such an approach would be reliable.

---

> ### Author Response · Authors · 2026-01-11
> **Response to Review**
>
> Thank you for taking the time to review our submission and provide thoughtful feedback.
>
> Possibility of Data leakage: We agree with the reviewer that this is a notable concern and, as the reviewer mentioned, it was already discussed in our initial draft. We will add text to elaborate on that discussion in the draft as follows. It is true that the models training data window includes the publication date of the raw survey dataset in its full granularity, which raises the possibility that there may be aggregated results or related analyses within the model’s training data. However, we perform our own aggregation of the survey data into demographic bins upon which all our predictive analysis is based. No prior publication of this data includes this aggregation, so there will be no direct leakage of the aggregate statistics we use to evaluate the LLM in our analysis into the model training. We are not aware of any mechanism by which an LLM could memorize arbitrary aggregate statistics of a structured survey dataset, even if the raw survey data may have been previously published in a machine readable tabular format.
>
> Limited coverage of Models and Datasets: It is true that this particular study focuses primarily on a single model, and it is beyond the scope of our current work to do a comparative analysis across models. We note that we have performed our work with the most prevalently used model (i.e., ChatGPT) currently used in US-based social science research contexts, so it is a suitable focal point for this study. The polling methods and frameworks proposed in this study can directly be adapted to determine if other models are similarly suited for using in predicting political opinions and we will expand our discussion, which briefly touches on this point already, to  further highlight the importance of this as further work.
>
> Lack of Experiments with RAG-based LLMs: We agree that it is of significant interest to analyze RAG-based LLM systems, although it is beyond the scope of this work. We demonstrate that even closed LLM setups perform well in predicting the responses of human political polls. While it is true that LLMs are unable to be trained on political content that is completely up-to-date, the timescales required for traditional political polls mean that by the time results are collected, they may similarly be months behind the status quo. An LLM whose training cutoff lies a few months in the past may serve to provide useful insights to policy makers who are otherwise unable to afford or execute traditional polling techniques. In addition, due to the temporal lag of traditional human polls, it would be difficult to find an accurate human baseline by which to judge an LLM’s ability to reflect such political opinions. We will add a paragraph to the conclusion outlining the possibility of measuring the relative improvement achieved by adding up to date context via RAG and noting these considerations.

---

### Review · Reviewer_pBwU · 2025-12-28

**Summary Of Contributions:**

__Summary__

This paper studies whether LLMs can accurately reproduce distributional patterns of public opinion on political issues. Using 84 multiple-choice questions from the 2022 Cooperative Election Study (CES) and 20 demographic groupings, the authors propose a Direct Distribution (DD) querying framework that prompts an LLM to output the entire response distribution for a given demographic, rather than repeatedly simulating individual respondents. Through a systematic comparison with the standard Single Individual (SI) querying approach, the paper shows that DD querying yields more accurate and less overconfident estimates of human response distributions, at substantially lower computational cost. The authors further demonstrate that DD performance varies in structured, partially predictable ways across demographics and issue domains, enabling the use of regression models to predict LLM reliability prior to deployment.

__Strengths__
1) __Methodological contribution__:
The Direct Distribution querying framework is a clear and well-motivated alternative to existing “silicon sampling” approaches, and is convincingly shown to outperform repeated individual querying for distributional opinion estimation.
2) __Multi-metric analysis__:
The use of complementary metrics (NEMD, MD, and SDD) allows the authors to disentangle overall distributional mismatch from biases in mean sentiment and misestimation of response heterogeneity, leading to a nuanced and informative evaluation.
3) __Cost-awareness and scalability__:
The paper explicitly accounts for query efficiency and demonstrates that the proposed method achieves higher accuracy with far fewer LLM calls, which is critical for any realistic polling or social-science application.

__Weaknesses__:
1) __Conceptual limitations of LLM-based polling__:
The paper essentially motivates LLM-based polling by appealing to the models’ exposure to large volumes of online political content, but this neglects fundamental conceptual issues. Political opinions are time-dependent, and the framework does not address temporal alignment between the opinions encoded in the model and the target polling period. Moreover, online data over-represent specific demographics, are affected by coordinated behavior (e.g., bots, trolls), and include misinformation and partisan amplification. These factors raise concerns about whether LLMs can faithfully approximate population-level opinion distributions, independent of prompting strategy, which are not addressed in the paper.
2) __Single-model evaluation__:
All experiments are conducted using a single LLM (GPT-4o-mini). Given well-documented political and cultural biases in LLMs [1], a broader comparison across models trained by different organizations and in different geopolitical contexts (e.g., US-centric models vs. European models such as Mistral, or Chinese models such as Qwen) would be essential for drawing robust conclusions about the general viability of AI-based polling.
3) __Potential data leakage concerns__:
While the authors acknowledge possible data leakage, the concern remains severe. Because the CES 2022 data predate the model’s training cutoff, it is impossible to fully rule out that aggregated results (or closely related analyses) were present in the training corpus. The only reliable way to eliminate this concern would be to evaluate on polling data collected strictly after the model’s knowledge cutoff.
4) __Overly coarse demographic bins__:
The demographic categorization is extremely coarse, particularly for race/ethnicity (e.g., White vs. Non-white). While this simplification improves statistical stability, it substantially limits interpretability and risks obscuring meaningful heterogeneity. At this level of aggregation, it is challenging to draw actionable conclusions about representational biases or subgroup performance.
5) __Restricted question format__:
The analysis is limited to multiple-choice Likert-style questions. It remains unclear how the proposed framework would extend to open-ended questions, multi-issue tradeoffs, or survey designs that more closely resemble real-world political polling instruments.

[1] Bang et al. 2024, Measuring Political Bias in Large Language Models: What Is Said and How It Is Said

**Audience:**

Yes

**Audience Explanation:**

The paper would be of interest to a subset of TMLR’s audience, particularly researchers working on the evaluation and use of large language models in social science settings, survey methodology, and distributional prediction. Its analysis of alternative prompting strategies for eliciting population-level response distributions, as well as the finding that such performance can be partially predicted from question and demographic features, offers methodological insights that are relevant beyond the specific political polling application.

**Broader Impact Concerns:**

This work raises __non-trivial ethical and societal concerns that are not sufficiently addressed in the current submission__ and would warrant a clearer and more explicit Broader Impact discussion.

A primary concern is the potential for skewing political opinion and influencing democratic processes. By framing LLM-generated distributions as approximations of public opinion, the proposed methodology risks being interpreted, or misused, as a substitute for human polling. Given that LLMs are trained on online data that systematically over-represent certain demographic groups, political ideologies, and modes of expression, the resulting “synthetic polls” may amplify existing biases rather than reflect the true distribution of public sentiment. Deploying such estimates in real-world political contexts (e.g., campaigns, policy design, media reporting) could __distort perceptions of public consensus and, in turn, influence voter behavior or policymaking__.

A closely related issue is the __temporal misalignment of political opinions__. Political attitudes are dynamic and can shift rapidly in response to events, yet LLMs encode opinions aggregated over long and opaque time horizons. Without explicit temporal conditioning, AI-generated polling outputs may reflect outdated or historically dominant views, potentially mischaracterizing current public opinion. This is particularly problematic for highly salient or volatile issues, where inaccurate representations could meaningfully affect public discourse and democratic decision-making.

The paper also does __not sufficiently engage with the risk posed by misinformation, coordinated manipulation, and trolling in online political content__. Because LLMs are trained on data that may include propaganda or artificially amplified viewpoints, their outputs may inadvertently legitimize or normalize distorted narratives when presented as population-level opinion estimates.

Finally, the use of __coarse demographic categories__ raises concerns about erasure and misrepresentation. Overly aggregated groupings, especially along racial or ethnic lines, may mask meaningful differences within groups and could be misused to justify oversimplified or misleading claims about the political preferences of marginalized populations.

Given these risks, a Broader Impact Statement should clearly articulate that LLM-based polling is not a neutral measurement tool, discuss safeguards against misuse, and emphasize that such systems should augment (rather than replace) rigorous, transparent human-centered polling in democratic contexts.

**Claims And Evidence:**

No

**Claims Explanation:**

While the paper provides solid empirical evidence for its methodological comparison, showing that the Direct Distribution framework outperforms single-individual querying under the studied setup, the broader claims about the suitability of LLMs for political polling are not fully supported. The evaluation relies on a single model, uses survey data that predates the model’s knowledge cutoff (raising unresolved data leakage concerns), and does not address temporal alignment of political opinions or known representation and bias issues in online data. As a result, the evidence is insufficient to substantiate the paper’s broader motivating claims.

**Requested Changes:**

1) __Address temporal alignment of political opinions (Critical)__:
The submission does not address the temporal nature of public opinion, despite motivating LLM-based polling by the models’ exposure to large volumes of online political discourse. The authors should explicitly discuss the temporal mismatch between LLM training data and the target polling period, and ideally evaluate the framework on polling data collected after the model’s knowledge cutoff, or introduce a time-aware analysis (e.g., by comparing issues with known temporal volatility). At a minimum, this limitation should be clearly acknowledged and its implications discussed.
2) __Mitigate or rule out data leakage (Critical)__:
Although the paper acknowledges possible data leakage, this concern remains unresolved and materially affects the validity of the results. A stronger mitigation strategy is needed, such as an evaluation on post-cutoff survey data or an explicit robustness analysis demonstrating that results are unlikely to be driven by memorization of survey summaries. Without this, it is difficult to interpret the reported performance as genuine generalization.
3) __Evaluate multiple LLMs with different cultural and political biases (Critical)__:
The conclusions are based on a single LLM. Given well-documented political and cultural biases in LLMs, the authors should evaluate at least one additional model trained by a different organization and/or in a different geopolitical context (e.g., European or Chinese models). This is essential to assess whether the observed advantages of the Direct Distribution framework generalize beyond a single model.
4) __Discuss the conceptual limitations of LLM-based polling more explicitly (Critical)__:
The paper should more directly engage with fundamental limitations of using LLMs for political polling, including over- and under-representation of demographic groups in online data, susceptibility to misinformation and coordinated behavior, and the distinction between modeling expressed online discourse versus true population opinion. A clearer conceptual framing would help prevent overinterpretation of the results.
5) __Refine or expand demographic categorization (Strengthening)__:
The current demographic bins (particularly the binary race categorization) are extremely coarse, making the results difficult to interpret. Where feasible, the authors should explore finer-grained groupings or provide additional justification and sensitivity analysis showing that the conclusions are robust to alternative demographic definitions.
6) __Extend beyond multiple-choice Likert-style questions (Strengthening)__:
The evaluation is limited to fixed-choice survey questions. Exploring whether the Direct Distribution framework can be applied to open-ended questions, multi-issue tradeoffs, or more realistic polling instruments would substantially strengthen the contribution and broaden its relevance.
7) __Clarify scope and claims (Strengthening)__:
The authors should more clearly distinguish between claims that are empirically supported (method-level improvements over SI querying) and those that are speculative (broader applicability of LLMs for political polling). Tightening this distinction would improve clarity and align claims more closely with the presented evidence.

---

> ### Author Response · Authors · 2026-01-11
> **Response to Review (Requested Changes 1-5)**
>
> Thank you for taking the time to review our submission and provide thoughtful feedback.
>
> 1 ) Addressing temporal alignment of political opinions and 4 ) discussing conceptual limitations: We agree with the conceptual limitations outlined by the reviewer and share the view that it remains to be seen if LLMs can usefully approximate public opinion to the extent of being useful in social science and campaign contexts; this work is meant to advance that characterization and the framework for such an evaluation. We are adding discussion in the paper to note the particular limitations raised by the reviewer about time dependency, representation, and coordinated / inauthentic behavior. As the reviewer suggested, we will add a paragraph explicitly discussing the temporal alignment limitations reflected by the model’s knowledge training cutoff. We do also want to note here that these are not conceptual limitations specific to the use of AI in polling, but rather are issues that affect public opinion research broadly, and require study as to their particular manifestation with new research methods like AI. For example, while it is true that LLMs are unable to be trained on political content that is completely up-to-date, the timescales required for traditional political polls mean that by the time results are collected, they may similarly be months behind the status quo. An LLM whose training cutoff lies a few months in the past may serve to provide useful insights to policy makers who are otherwise unable to afford or execute traditional polling techniques. Furthermore, while it is the case that coordinated bots and trolls may cause opinions present on the internet to be skewed, traditional polling methods are also vulnerable to inauthentic responses associated with partisanship, trolling, etc. (see for example https://papers.ssrn.com/sol3/papers.cfm?abstract_id=3131087). Our findings help constrain the relative impact of effects like these on AI-based public opinion measurement, relevant to our ground truth sample of traditional polling data. We are adding a paragraph to the paper explicating the nature of these tradeoffs and the role of our contribution in characterizing them.
>
> 2 ) Mitigating Data leakage: We agree with the reviewer that this is a notable concern and, as the reviewer mentioned, it was already discussed in our initial draft. We will add text to elaborate on that discussion in the draft as follows. It is true that the models training data window includes the publication date of the raw survey dataset in its full granularity, which raises the possibility that there may be aggregated results or related analyses within the model’s training data. However, we perform our own aggregation of the survey data into demographic bins upon which all our predictive analysis is based. No prior publication of this data includes this aggregation, so there will be no direct leakage of the aggregate statistics we use to evaluate the LLM in our analysis into the model training. We are not aware of any mechanism by which an LLM could memorize arbitrary aggregate statistics of a structured survey dataset, even if the raw survey data may have been previously published in a machine readable tabular format.
>
> 3 ) Evaluating multiple LLMs: It is true that this particular study focuses primarily on a single model, and it is beyond the scope of our current work to do a comparative analysis across models. We note that we have performed our work with the most prevalently used model (i.e., ChatGPT) currently used in US-based social science research contexts, so it is a suitable focal point for this study. The polling methods and frameworks proposed in this study can directly be adapted to determine if other models are similarly suited for using in predicting political opinions and we will expand our discussion, which briefly touches on this point already, to  further highlight the importance of this as further work.
>
> 5 ) Addressing overly-coarse demographic categorizations: as identified by the comment, further detailed demographic bins would reduce the statistical stability of comparisons to the baseline. We have already evaluated the tradeoff between sample size and bin coarseness in order to reach the categorization used in the paper. We will add a paragraph discussing these considerations to the paper.

---

> > ### Author Response · Authors · 2026-01-11
> > **Response to Review (Requested Changes 6-7 and Broader Impact)**
> >
> > 6 ) Addressing restricted question format: the analysis is purposefully limited to binary, categorical, and ordinal (including “likert-style”) questions. Large-scale real-world political polling instruments generally  focus on structured response questions given the ease of aggregating responses and the low effort required of respondents, making them highly suitable to large respondent populations. We will add a paragraph explicitly noting this limitation and calling out the possibility of future work to assess the applicability of LLMs to other question formats, such as open ended responses.
> >
> > 7 ) Clarifying scope and claims: We will revisit claims made throughout the paper and make textual edits where necessary to clarify what claims are and are not based on data presented in the paper.  We will revise claims such as “[AI-based polling techniques] can open the benefits of political issue polling to elected representatives, candidates, and jurisdictions that do not have the resources to field polls of human respondents.” to more clearly draw attention to the difference between the scope of the paper and broader potential implications of political polling, as well as address broader impact concerns. We assess that only a small number of claims will require modification, as the current draft generally does make clear what specific quantitative data from our study or the cited literature underlies each claim.
> >
> > Addressing broader impact concerns: We will follow the reviewer’s suggestion to add a broader impact statement emphasizing that LLM-based polling is not a neutral measurement tool, discuss safeguards against misuse, and emphasize that such systems should augment (rather than replace) human polling. We want to emphasize here, as we did already in the draft itself, that our intent is not to recommend replacing traditional human political polls when the infrastructure for such exists. Rather, we are grappling with a fast emerging reality that pollsters, scientists, and campaigns are experimenting with AI based tools to measure public opinion and the gap in understanding about the relative accuracy and appropriateness for use of these tools. Our view is that it is critical to identify whether and under what conditions LLMs are suitable for predicting political polling results, and this study proposes both a novel polling method and a framework for evaluating the accuracy of LLM polling predictions, making it a useful contribution to the literature, where a method for determining the validity of LLM polling predictions is of great interest.

---

> > > ### Comment · Reviewer_pBwU · 2026-01-13
> > >
> > > Thank you for taking the time to provide such a detailed and thoughtful rebuttal. I appreciate the care with which you engaged with the review and the effort to clarify scope, limitations, and broader impact considerations.
> > >
> > > That said, even with the proposed revisions, my main concern remains that the paper lacks sufficient depth from a technical and machine learning perspective. The reliance on a single model, unresolved data leakage and temporal validity issues, and the absence of cross-model or time-aware empirical analysis limit the strength and generality of the conclusions. While the work offers useful methodological insights and careful characterization, these limitations ultimately prevent it from meeting the level of technical rigor I would expect for a stronger recommendation.

---

### Decision · Action_Editor_Sc1D · 2026-02-12

**Recommendation:** Reject

**Audience:**

Yes

**Audience Explanation:**

Although the reviewers did say that this is of some interest, there were some concerns raised about the fitness for TMLR. Two reviewers categorized this work as being a bitter fit with a data science or applied statistics venue.

**Claims And Evidence:**

No

**Claims Explanation:**

All reviewers flagged a major concern: the ground truth dataset predates the training cutoff for the evaluated LLMs. This makes it challenging to interpret the results and undermines the claims. There were a number of other more minor points, but ultimately, after rebuttal and discussion, the reviewer unanimously recommended rejection.